ම | **Open Peer Review** | Parasitology | Research Article

# Exploring virulence and stress response in *Entamoeba histolytica*: insights from clinical strains

Yasuaki Yanagawa,[1,2] Manu Sharma,[1] Shinji Izumiyama,[3] Upinder Singh[1,4]

**ABSTRACT** Amebiasis, caused by the protozoan *Entamoeba histolytica*, is a significant parasitic infection affecting millions of people worldwide, particularly in developing regions. Elucidating the mechanisms by which this parasite invades host tissues and circumvents immune defenses is essential for advancing therapeutic strategies. Key virulence factors such as Gal/GalNAc lectin, amebapore, and proteases enable the parasite to adhere to, invade, and destroy host tissues. This study aimed to identify novel virulence-related genes in clinical *E. histolytica* strains by analyzing their gene expression profiles. We collected clinical isolates from asymptomatic individuals and patients with amoebic liver abscesses to perform RNA sequencing and compare their gene expression profiles. The analysis identified 14 differentially expressed genes between high-virulence and low-virulence strains. Among these, four candidate genes exhibited significant upregulation in the virulent strains. Functional assays demonstrated that the overexpression of these genes contributed to key virulence traits, including increased adhesion, complement resistance, and enhanced starch phagocytosis. Significantly, two of the four candidate genes, EHI_124550 and EHI_107170, which encode hypothetical proteins, exhibited a strong correlation with both oxidative stress response and complement resistance. These findings suggest that specific genes play crucial roles in the parasite's ability to evade the host immune system and establish infection in extraintestinal sites like the liver.

**IMPORTANCE** This study focuses on understanding how *Entamoeba histolytica*, the parasite responsible for amebiasis, affects over 50 million people globally. Our research is the first study to examine various clinical strains of the parasite, identifying key genes that influence its ability to attach to host cells (adhesion) and ingest them (phagocytosis), both critical processes for its ability to cause disease. Additionally, we discovered that these genes play a role in helping the parasite withstand environmental stress, such as oxidative stress and heat shock, which are part of the body's defense mechanisms. These findings are significant because they reveal potential targets for future treatments aimed at reducing the parasite's virulence, or disease-causing potential. Understanding how *E. histolytica* adapts and survives under hostile conditions will help in developing better strategies to combat amebiasis. These results provide new insights into a unique immune evasion strategy employed by a pathogen.

**KEYWORDS** *Entamoeba histolytica*, liver abscess, transcriptome analysis, virulence gene, oxidative stress, adhesion, phagocytosis, GRIP

Amebiasis refers to an invasive intestinal or extraintestinal infection caused by the protozoan *Entamoeba histolytica*. It affects over 50 million people globally, with up to 110,000 deaths each year (1–3). In terms of parasitic fatalities, only malaria and schistosomiasis rank higher than amebiasis (3, 4). As travel and immigration to developed countries rise, infections are becoming more frequent in non-endemic regions.

Address correspondence to Upinder Singh, usingh@stanford.edu.

The authors declare no conflict of interest.

See the funding table on p. 15.

While the majority of *E. histolytica* infections remain asymptomatic, some individuals develop amoebic colitis and systemic disease. Growing knowledge of its pathogenesis and immune response offers hope for future vaccine development (2). Amebiasis, caused by *E. histolytica*, begins with ingestion of cysts, leading to colonization of the large intestine (5). The parasite can invade intestinal mucosa, causing amoebic colitis or dysentery. In severe cases, it spreads to the liver via the bloodstream, resulting in amoebic liver abscesses, potentially causing systemic disease (6). Recent findings indicate that the coronavirus disease 2019 pandemic might have impacted the rise in amoebic liver abscess incidence through changes in the host immune response (7). *E. histolytica* is capable of invading human tissues through various molecules and biological mechanisms related to virulence (8–10). Pathogenic amoebas utilize three primary virulence factors—Gal/GalNAc lectin, amebapore, and proteases—to lyse, phagocytose, kill, and destroy a range of host cells and tissues (11–14). Recent investigations have improved our insight into the cell biology and gene regulation of *E. histolytica*. In the amoeba, dominant-negative mutations in the Gal/GalNAc lectin negatively impact adhesion and cytolysis (15). The gene library of the laboratory *E. histolytica* wild-type strain provides an extensive collection of genetic sequences for research purposes (16–18). Known for its virulence, this reference strain allows researchers to explore gene expression, regulatory pathways, and mechanisms of pathogenicity, advancing studies on parasite-host interactions and parasite biology. However, the gene expression profile of *E. histolytica* clinical strains differs significantly from that of the laboratory strains (19, 20). Clinical strains often display unique gene regulation patterns, particularly in virulence-related genes, which would reflect their adaptation to host environments, whereas the laboratory strain shows a more stable and less varied expression profile. Although there was a report to study the comprehensive transcription focusing on the pathogenic and non-pathogenic *E. histolytica* isolates, there are few reports on genetic analysis using libraries from various strains (21). The complexity and difficulty of culturing the parasite from clinical samples are the main factors contributing to this problem (22). The difficulty of cultivating trophozoite- and/or cyst-containing clinical samples is significant, and success is largely dependent on the time elapsed from collection to culture and on the skill of the microbiologist. Even with the obstacles involved, we have reported findings on the genetic characteristics of several clinical strains, made possible by the strong partnership between the clinical and research sectors (19). In this study, we aimed to identify novel genes that phenotypically influence the parasite's virulence using previously uncharacterized *E. histolytica* clinical strains.

## RESULTS

### Selection of candidate genes based on virulence in human and animal models

To evaluate whether the clinical strains of *E. histolytica* isolated from patients exhibit similar virulence traits in animal models, we collected specimens from both asymptomatic individuals and those with amoebic liver abscesses diagnosed by PCR. Building on our earlier work, where we established gene expression profiles using a few clinical strains (19), we have now broadened the study by gathering and culturing more isolates to identify shared candidate genes that directly impact the virulence of *E. histolytica* across different infection severities. As a result, we successfully established four axenically cultured isolates: two (NA11 and NA124) from the stool of asymptomatic individuals and two (NA19 and NA148) from liver abscess aspirates. In terms of virulence, the isolates from liver abscesses (high-virulence [HV] group) showed similar pathogenicity in hamster models (Fig. 1A). Conversely, the asymptomatic isolates (low-virulence [LV] group) failed to induce liver abscess lesions in animal models, in agreement with our earlier findings (19). This finding highlights the distinct *in vivo* virulence of clinical *E. histolytica* strains. To further investigate, we analyzed the RNA-seq profiles of three biological replicates of each strain to determine gene expression patterns under identical conditions. This analysis identified 12,375 transcripts, with differentially expressed genes

A.

| Group | Research ID | Clinical status in human | Liver abscess in animal model |
|---|---|---|---|
| High Virulent (HV) | NA19 | Liver abscess | yes |
| | NA148 | Liver abscess | yes |
| Low Virulent (LV) | NA11 | Asymptomatic | no |
| | NA124 | Asymptomatic | no |

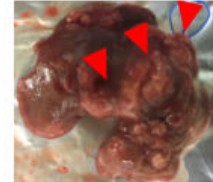

Liver abscess in hamster, NA148

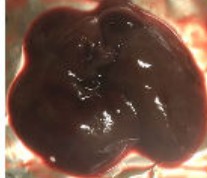

No liver abscess in hamster, NA124

B.

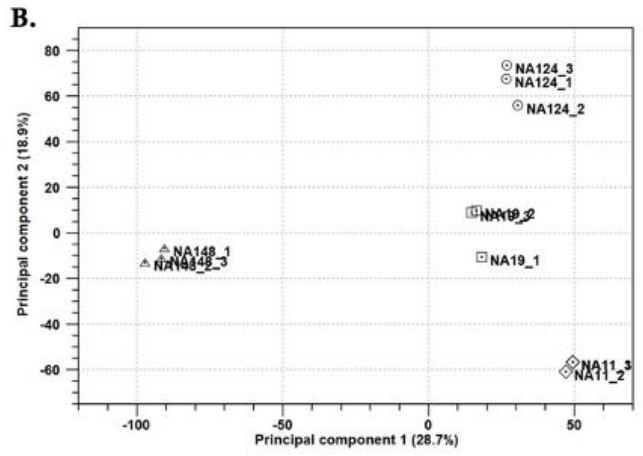

C.

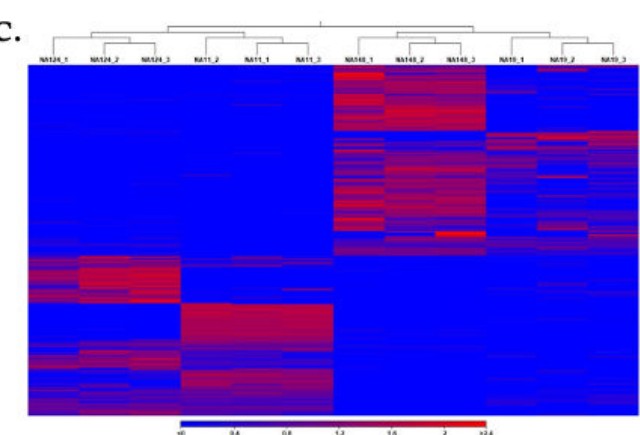

D.

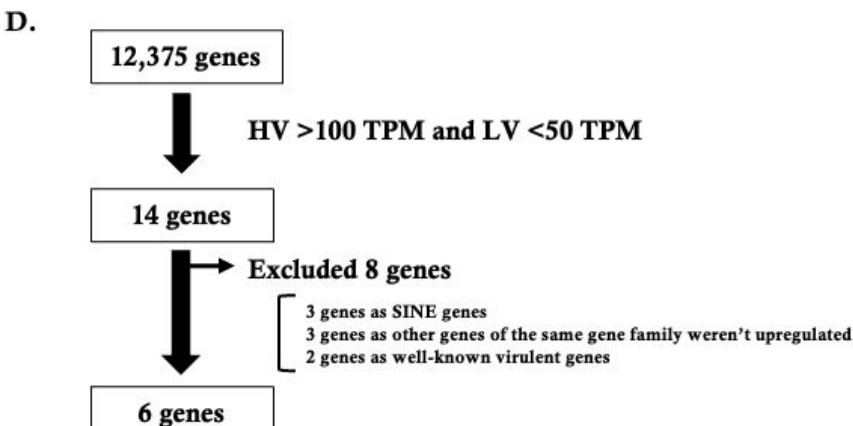

FIG 1 RNA-seq analysis of four clinical *E. histolytica* strains. (A) The four clinical *E. histolytica* strains were divided into two groups according to their ability to produce liver abscesses in an animal model. Representative images of animal experiments are shown for each strain in the high-virulence (HV) and low-virulence (LV) groups, with red arrowheads highlighting liver abscess lesions. (B) Principal component analysis plot displaying RNA-seq data from triplicate analyses of the four clinical strains. Each data point represents a biological replicate, illustrating distinct gene expression profiles for each strain. (C) Hierarchical clustering of the strains based on Spearman rank correlation. (D) Gene selection process based on expression thresholds (HV ≥100 TPM; LV ≤50 TPM). Out of 12,375 genes, 6 genes were selected after excluding 8 genes. SINE, short interspersed transposable element.

(DEGs) defined as those with a false discovery rate (FDR) of less than 5%. Principal component analysis (PCA) demonstrated both the reproducibility of the replicates and differentiation among strains (Fig. 1B). Interestingly, the RNA-seq data for each *E. histolytica* isolate differed from other samples within the same virulent group (HV or LV), suggesting unique RNA expression profiles under similar *in vitro* conditions. We then generated a heatmap of the DEGs to compare overall gene expression

trends. Hierarchical clustering revealed several genes that were more upregulated in HV strains than in LV strains (Fig. 1C). In terms of global gene expression, each clinical strain demonstrated a unique expression pattern. Finally, to identify genes specifically expressed in the virulent strains, we compared the gene expression profiles of HV group strains with those of LV group strains. By applying two different multiple comparison methods (FDR multiple ANOVA and Tukey's comparison analysis) and a cutoff of HV ≥100 transcripts per million (TPM) and LV ≤50 TPM, we identified 14 DEGs across the four strains (Fig. 1D). Out of these, eight genes were excluded from further study: three were classified as short interspersed transposable elements; three belonged to gene families not upregulated in other members; and two were well-known genes with established molecular functions (EHI_061980 and EHI_183460) (23–26).

## Features of the candidate genes that are upregulated in the virulent group

Of the six genes with elevated expression in the HV group, excluding the eight excluded genes, we focused on the top four candidates (Fig. 2A; Table 1). Comparative analysis of the TPM mean values between the two groups revealed no significant differences, aligning with the observed variations in pathogenicity. Consequently, we focused on the four genes with elevated expression levels in the HV group. The RNA-seq analysis revealed that the TPM count of U2 (EHI_C00154) in the HV group was 69 times higher than that in the LV group. The other three genes, U15 (EHI_124550), U16 (EHI_107170), and U18 (EHI_176850), exhibited a three- to fivefold difference in expression levels. According to the *E. histolytica* genome assembly database, these four genes are predicted to encode hypothetical proteins of unknown function. Among the four genes analyzed, U15, U16, and U18 exhibited high sequence homology in BLASTN searches for nucleotide sequence similarity, with $E$ values of 1e-177 for U15 against *Entamoeba dispar*, 3e-20 for U16 against *Entamoeba nuttalli*, and 0.0 for U18 against both *E. nuttalli* and *E. dispar*. BLASTP similarity searches showed that the proteins encoded by U2, U15, and U16 have no homology to organisms outside of the *Entamoeba* spp. (Fig. S1). These putative proteins are similar to those in *E. nuttali* (for instance, the U15 gene has 97.3% identity; $E$ value = 1e-97.3). The U18 gene's protein sequence is similar to those in other organisms, such as *Solanum pennellii* and *Sphagnum jensenii*. RNA-seq read mapping of the four identified genes revealed previously unannotated transcribed regions within the genome. We then translated their sequences and analyzed for functional protein domains using the Pfam database (27, 28). While the U2 and U16 genes did not show any significant protein domains, putative domains were identified in the U15 and U18 genes (Fig. S2). The U15 gene encodes a membrane-bound protein predicted to be located in the cytoplasm with no match to annotated proteins in other organisms. The U16 gene demonstrates notable homology with the surface antigen Ariel 1 of *E. histolytica* HM-1. This antigen is characterized by its asparagine-rich composition and features 2–16 octapeptide repeats, as reflected in its nomenclature and functional encoding (29). On the other hand, the U18 gene encodes a protein that contains a golgin-97, ranBP2α, imh1p, and p230/golgin-245 (GRIP) domain, a specific targeting sequence for the trans-Golgi network in animal cells (30). To further validate and compare gene expression between the laboratory virulent strain (wild type) and the avirulent strain (Rahman), we performed reverse transcription semi-quantitative PCR (Fig. 2B). The U2 gene was notably more upregulated, while the U16 and U18 genes also showed relatively higher expression levels. These findings generally reflect the gene expression levels observed in our RNA-seq data from clinical *E. histolytica* strains. To evaluate the biological significance of each candidate gene on parasite virulence, trophozoites were transfected to overexpress each target gene, fused to a Myc tag at the N-terminal end. The full length of each gene was obtained by PCR using genomic DNA as a template under the conditions described in the Materials and Methods. Transfected parasites were then selected in the presence of 24 μg/mL of G418. Western blot analysis with whole-cell lysates confirmed overexpression (Fig. 2C). Single bands were detected at the expected molecular weights, except for the U2 gene. In the U2-transfected parasite lane, a band of 20 kDa appeared,

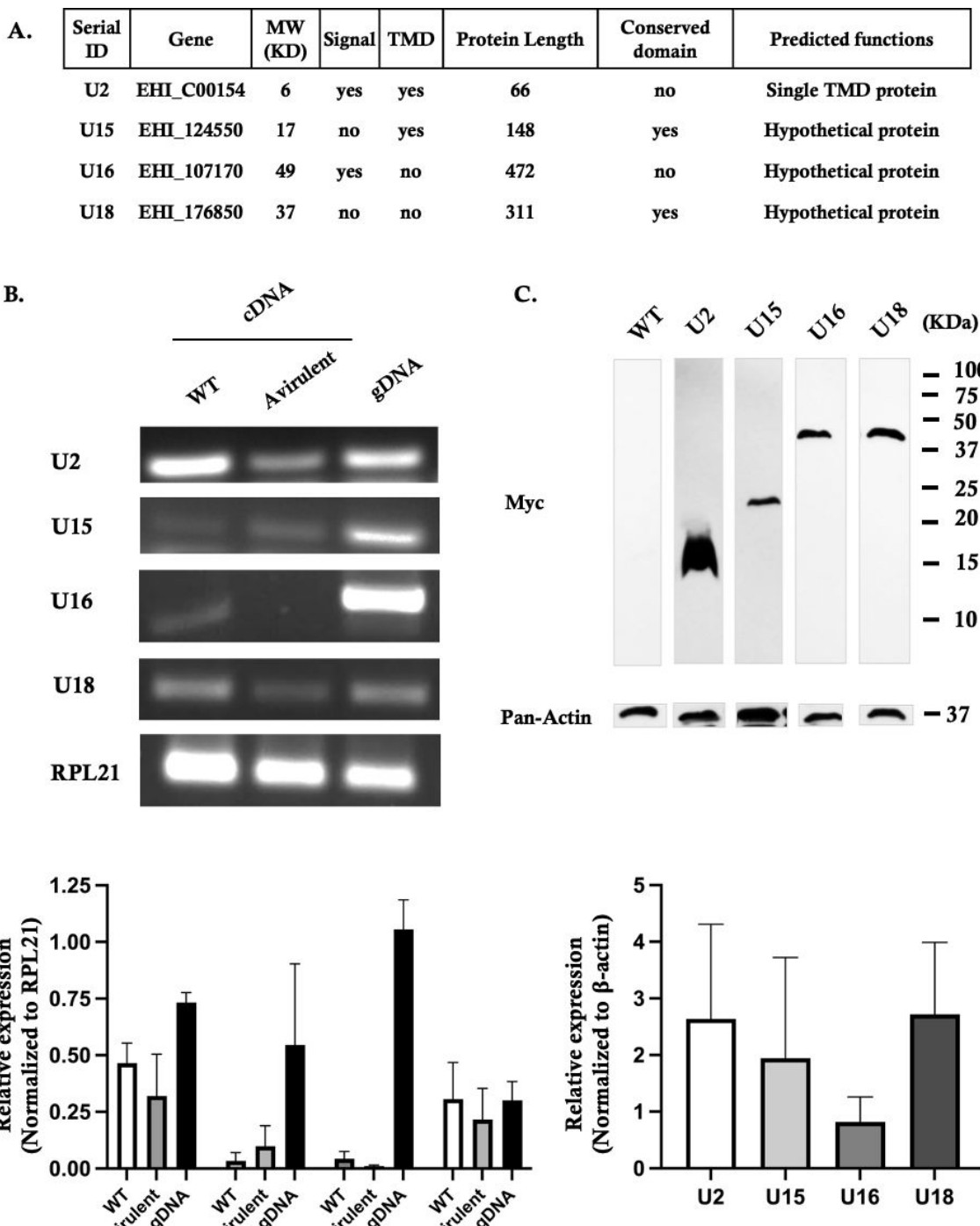

FIG 2 Overview of the four genes upregulated in the HV group. (A) List of the four selected genes, with information on molecular weight (MW) and the presence of transmembrane domains (TMDs). (B) Reverse transcription PCR analysis of the selected genes in the virulent wild-type (WT) strain and the avirulent strain. Genomic DNA (gDNA) from the wild-type trophozoites was used as a control, and ribosomal protein 21 (RPL21) served as a stable reference gene. (C) Western blot analysis of overexpressed Myc-tagged genes. Total lysates (approximately 20 µg) from the transfectants were separated on an SDS-PAGE gel. The lysate from WT was used as a negative control for Myc-tagged protein in this assay.

twice the expected size of 10 kDa. The 10 kDa expected size is calculated from the sum of the pKT3M plasmid (6 kDa) and the U2 protein (4 kDa). Despite efforts to denature

**TABLE 1** Genes differentially more highly expressed in the HV group than those in the LV group (threshold HV ≥100 TPM + LV ≤50 TPM, padj ≤0.05)[a]

| Serial ID | Gene | HV (TPM) | | LV (TPM) | | Fold change | FDR P value |
|---|---|---|---|---|---|---|---|
| | | NA19 | NA148 | NA11 | NA124 | | |
| U2 | EHI_C00154 | 410.9 | 8.1 | 12.8 | 5.4 | 69.1 | 0.021 |
| U15 | EHI_124550 | 44.2 | 226.8 | 17.2 | 31.1 | 5.61 | 5E-04 |
| U16 | EHI_107170 | 45.3 | 193.8 | 19.8 | 48.5 | 3.50 | 0.027 |
| U18 | EHI_176850 | 27.1 | 288.4 | 39.9 | 34.9 | 37.4 | 1E-04 |

[a]HV, high virulence; LV, low virulence; TPM, transcript per million.

the dimeric protein using reducing agents like beta-mercaptoethanol or urea, the band remained intact. Similarly, we sought to evaluate the target function via RNA interference silencing models previously described (31), but all transfectants failed to survive during G418 drug selection.

## Subcellular localization and stress response to environmental factors

We performed an immunofluorescence assay (IFA) to assess the localization of each target gene using Myc-tagged cell lines (Fig. 3A). All transfectants exhibited cytoplasmic signals. U15 and U16 showed a punctate, pigment-like pattern within the cytoplasm. The doubling times of the transfectants were assessed, showing similar growth rates to the mock-pKT3M strain (U2, 21.96 h; U15, 23.48 h; U16, 19.61 h; U18, 24.48 h; mock, 21.13 h). Given that the activation of stress responses is crucial for adaptation to environmental challenges, we evaluated the parasite's responses to heat shock, oxidative stress, and human complement exposure. The oxidative stress assay revealed that U15 and U16 cell lines were resistant to oxidative stress (U15: 1.53-fold, U16: 1.56-fold; both $P < 0.05$), whereas the U18 cell line was more sensitive than the control (0.51-fold, $P < 0.05$) (Fig. 3B). Similarly, the U15 and U16 cell lines revealed resistance to heat shock (U15: 1.63-fold, U16: 1.41-fold; both $P < 0.01$) (Fig. 3C). *E. histolytica* has developed various strategies to resist the human complement system, including surface molecules that inhibit complement activation, shedding of complement-bound membranes, and disrupting the complement cascade (32). To investigate a biologically relevant phenotype related to complement resistance, we tested the transfectants' resistance to lysis by human complement for 1 h. Consistent with the results of the above two assays, the U15 and U16 cell lines exhibited increased resistance to complement stress (U15: 1.35-fold, $P < 0.05$; U16: 1.45-fold; both $P < 0.01$), while the U2 cell line was more susceptible (0.44-fold, $P < 0.01$) (Fig. 3D). These data suggest that the U15 and U16 genes may be involved in the mechanism of complement resistance.

## Overexpression of the candidate genes elevates adherence and phagocytosis levels

Once *E. histolytica* colonizes the human large bowel, it can sometimes penetrate the intestinal mucosa and spread to other organs. The liver is the most affected organ, reached through the bloodstream via the portal vein from the colon. During this process, the parasite is able to evade host defense mechanisms to ensure its survival. Since our HV strains can induce liver abscesses in both humans and hamsters, we hypothesized that they possess unique biological traits that allow them to resist the human immune response. To explore the biological functions of the four candidate genes, we selected a broad range of phenotypic assays, including adhesion, monolayer destruction, starch phagocytosis, and erythrophagocytosis.

Surface molecules like Gal/GalNAc lectin and KERP1 regulate adhesion, signaling, ingestion, and immune modulation at the host-parasite interface (9, 11, 34). Since the killing of target cells by *E. histolytica* is primarily contact dependent, a reduction in its ability to adhere to target cells should lead to a decrease in cell death (35). We evaluated the adherence capacity of each transfectant. As illustrated in Fig. 4A, all transfectants

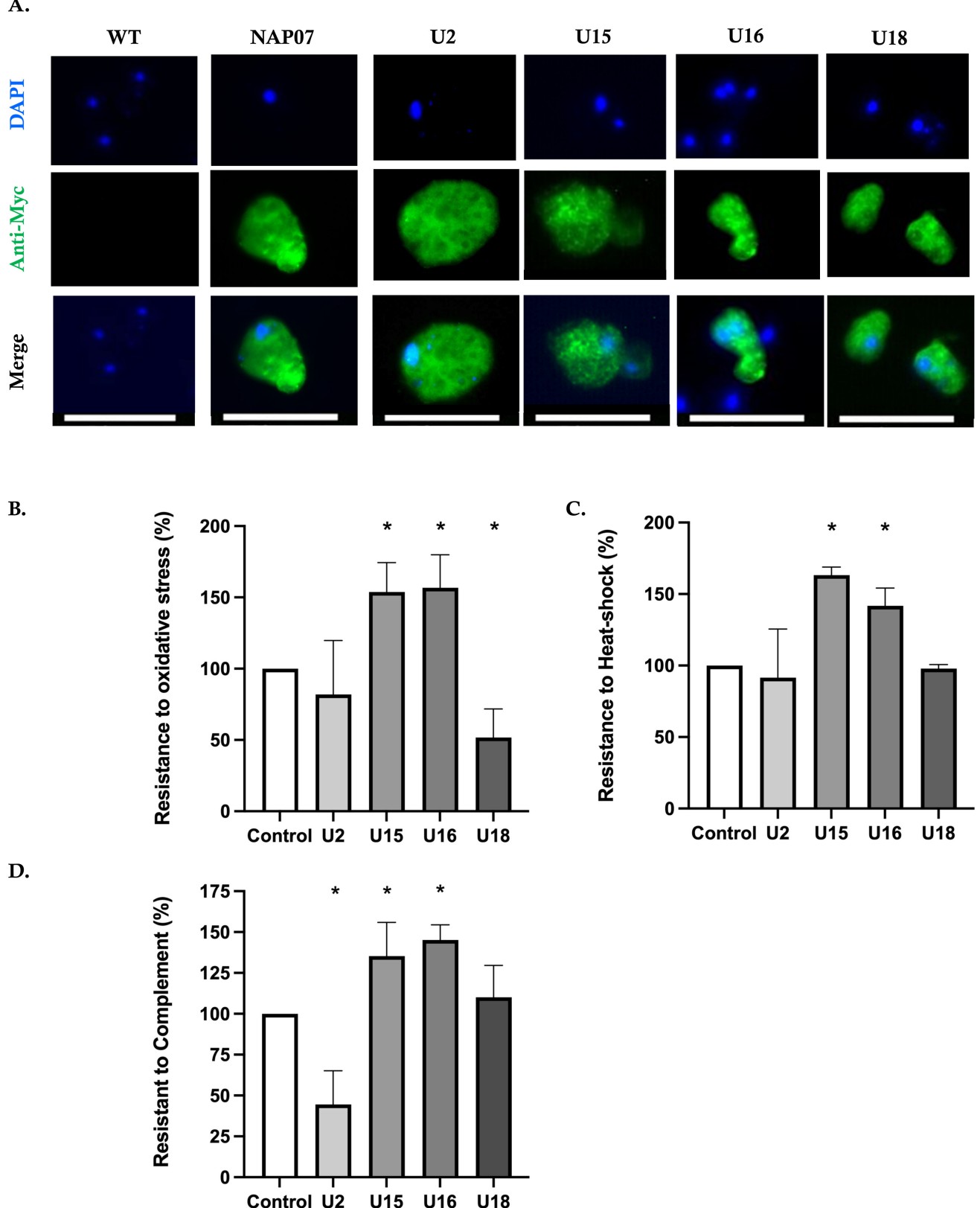

**FIG 3** Subcellular localization of the four genes in *E. histolytica* and their impact on stress resistance. (A) Localization of Myc-tagged overexpressing cell lines, showing cytoplasmic distribution of all four genes. Anti-pan actin was used as a control for loading. White bars, 50 µm. The trophozoite of WT was stained as a negative control, while that of NAP07 (nucleosome assembly protein), provided by Hanbang Zhang, was used as a positive control for the Myc-tagged protein (Continued on next page)

**Fig 3 (Continued)**

(33). (B) Oxidative stress viability was measured after exposing trophozoites to 2.5 mM $H_2O_2$ using trypan blue exclusion. Results were compared with control cells expressing CS-Luc plasmid under similar G418 conditions. (C) Heat-shock viability assay at 42°C for 3 h using the CS-Luc stable cell line as the control. (D) Complement resistance was assessed by incubating $1 \times 10^4$ trophozoites with 50% normal human serum (NHS) for 1 h at 37°C. Heat-inactivated NHS served as a control, and cell viability was determined by 0.2% trypan blue staining. Data represent the mean ± standard deviation of three independent experiments, conducted in duplicate. Student's $t$-test. *$P < 0.05$. DAPI, 4′,6-diamidino-2-phenylindole.

showed an increase in adherence (1.5- to 2.1-fold, $P < 0.05$). Similarly, there was an increase in the number of parasites with ≥4 attached Chinese hamster ovary (CHO) cells. We then tested whether these highly adherent transfectants would lead to increased virulence in the CHO cell monolayer destruction assay. Overexpressed transfectants ($6 \times 10^4$ cells) were incubated on a CHO cell monolayer ($3 \times 10^5$ cells) in a 24-well plate prepared 1 day prior to the assay. Unexpectedly, the monolayer destruction assay showed no significant change in virulence for U15 and U16 or a decrease in virulence for U2 and U18 compared to the control transfectant in their ability to disrupt the CHO cell monolayer (Fig. 4B). These findings were inconsistent with the results obtained from the stress response assays.

*E. histolytica* is known to phagocytose multiple types of cells and particles, including starch grains, bacteria, protozoa, and erythrocytes (36). Phagocytosis of starch and human red blood cells (RBCs) was examined in the overexpressing transfectants (Fig. 4C and D). Compared to the control, all four overexpressing transfectants showed an increase in starch phagocytosis (1.5- to 2.5-fold, $P < 0.001$), in line with their enhanced adherence to CHO cells. Interestingly, no changes were observed in erythrophagocytosis. Contrary to these findings, the rhomboid protease knockdown model exhibited reduced phagocytosis of both targets (37). The process of starch phagocytosis entails recognizing non-cellular particles, possibly influenced by specific receptors that bind to carbohydrates and other surface molecules. This indicates that distinct endocytic pathways could be involved, unlike those used for cellular targets such as RBCs.

## DISCUSSION

In this study, we identified distinct gene expression profiles in clinical strains of *E. histolytica* with high-virulence potential using RNA-seq analysis. Following the release of the *E. histolytica* genome sequence and resources in 2005, a number of genome-wide microarray gene expression profiling studies and RNA-seq analyses were performed under various conditions to gain deeper insights into the organism's biology (16–18, 27, 38–40). Transcriptomic studies have facilitated *E. histolytica* gene expression analysis under tissue invasion and environmental stress conditions (41, 42). These studies were key in differentiating gene expression between virulent and avirulent strains, revealing pathogenic mechanisms (43–45). Transcriptomic analysis with microarray revealed extensive differences between the laboratory virulent and avirulent strains (21). The wild-type *E. histolytica* showed high expression of virulence factors like cysteine proteases (CPs), Gal/GalNAc lectin, and lysozyme with a cecropin domain. AIG1 gene family members involved in bacterial resistance of *Arabidopsis thaliana* suggest that multiple pathways, including signal transduction and cytoskeletal rearrangements, influence virulence (46). Transcriptomic data provide insights into transcriptional regulation, mapping essential gene networks for survival and host adaptation (40). Our findings reveal several upregulated genes that may contribute to the parasite's ability to evade host immune responses and cause liver abscesses in both human and animal models. The failure to generate silencing transfectants indicates that these genes may be critical for *E. histolytica's* survival. Certain genes did not correspond to the previously identified virulence-associated genes that were upregulated in virulent strains, as a large portion of *E. histolytica* genes remain to be annotated with unknown functions. In this context, several prominent virulence genes, including cysteine protease, peroxiredoxin, and elongation factor-1 alpha, which are crucial for trophozoite survival under oxidative

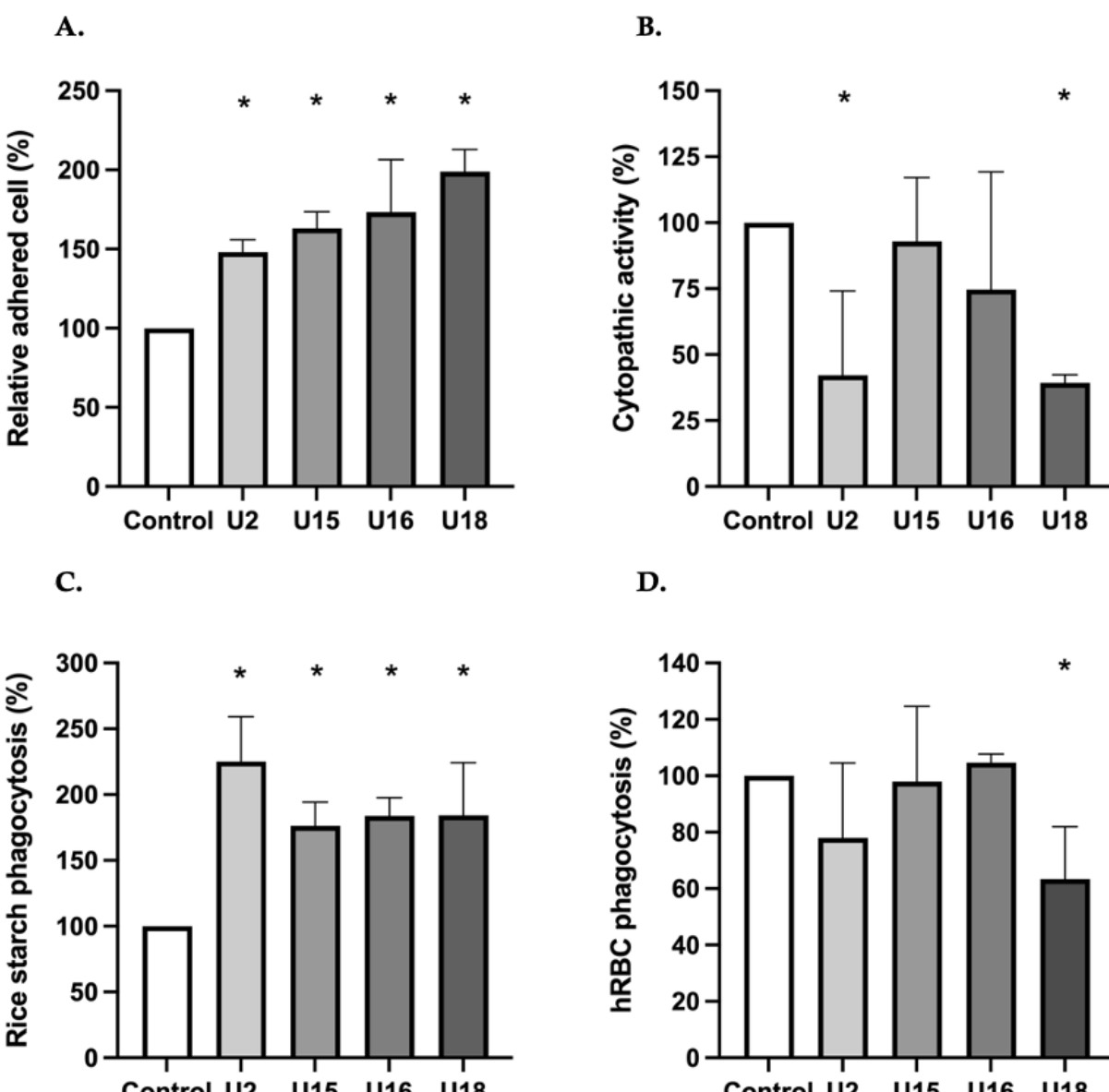

**FIG 4** Overexpression of four genes in *E. histolytica* results in increased phenotypic characters of adhesion and phagocytosis for rice starch. (A) Adhesion was measured with a CHO cell rosette assay. The trophozoites were combined with CHO cells at a 1:20 ratio and incubated on ice for 2 h. The amebae were washed twice in 1× phosphate-buffered saline, and parasites with ≥3 CHO cells attached were considered positive for adhesion. (B) Cytopathic activity was measured by placing a total of $6 \times 10^4$ trophozoites on a confluent CHO cell monolayer for 2 h at 37°C, followed by fixation with 4% paraformaldehyde, staining with 0.1% methylene blue, dye extraction, and spectrophotometric determinations at 650 nm. CHO cells were incubated with only medium and without amoebae, with 0% destruction. (C) A total of $1 \times 10^5$ trophozoites were incubated for 1 h with 0.004% rice starch solution followed by fixation, permeabilization, and staining with 1% Lugol's solution at room temperature for 5 min. Parasites with one or more ingested starch grains were considered positive for rice starch phagocytosis. (D) Erythrophagocytosis of *E. histolytica* trophozoites transfected with four genes. Transfected amoebae of the wild-type isolate were incubated with human erythrocytes in a ratio of 1:50 for 30 min. Average numbers of engulfed erythrocytes were quantified by measuring the absorbance at 405 nm after resuspending the pellet of washed trophozoites in 90% formic acid. Results were compared with control cells expressing CS-Luc plasmid under similar G418 conditions. Data represent the mean ± standard deviation of three independent experiments, conducted in duplicate. Student's *t*-test. *$P < 0.05$.

stress and the pathogenesis of amoebiasis, were selected from our transcriptome data set (24, 32, 47). This observation underscores that our transcriptome data not only capture previously established virulence-associated genes but also reveal potentially novel virulence determinants.

Of the genome sequence of *E. histolytica*, half is predicted to be coding sequence, 8,333 annotated genes (48). However, the assembly of 1,496 scaffolds is considered incomplete as it may contain misassembled regions and incorrectly annotated gene models (38). This could be explained by the unique finding that homologs for approximately 31.8% of predicted *E. histolytica* proteins remain unidentified in public databases (17).

Environmental stress triggers defensive responses, activating protective mechanisms for survival. Heat and oxidative stress studies offer crucial insights into eukaryotic cell biology (49). To survive in the host, the parasite adapts to environmental pressures, enabling immune evasion and promoting its sustained survival (50). Oxidative stress and heat-shock responses are critical for *E. histolytica* survival in hostile environments, particularly within the host, where immune cells like macrophages and neutrophils generate reactive oxygen species (ROS) (51). The parasite activates antioxidant defenses like superoxide dismutase and peroxiredoxin to neutralize ROS and maintain cellular integrity (52, 53). Heat-shock proteins function as chaperones, ensuring proper protein folding and repair during elevated temperature or metabolic stress conditions (54). These oxidative stress and heat-shock responses enable *E. histolytica* to evade immune destruction, enhancing its adaptability and persistence in diverse host environments. Overexpression of *E. histolytica* Dnmt2 homolog (Ehmeth) enhances EhHsp70 expression, improving trophozoite resistance to hydrogen peroxide ($H_2O_2$)-induced oxidative stress, suggesting a potential epigenetic regulation of EhHsp70 in oxidative stress defense (55). In our study, both genes (EHI_124550 and EHI_107170) exhibited resistance to oxidative and heat-shock stress, despite undefined functional domains. These upregulated genes in highly virulent *E. histolytica* strains likely contribute to stress response mechanisms, possibly enhancing virulence. This correlates with the wild-type strain's virulent phenotype and its superior ability to reduce $O_2$ and $H_2O_2$ (56). In addition, *E. histolytica*'s evasion of the host complement system is vital for survival, particularly during blood flow to the liver. The parasite inhibits complement activation, degrades complement proteins, and sheds complement-bound fragments, preventing cell lysis and promoting virulence and tissue invasion (57). Recent studies also suggest that *E. histolytica* evades complement lysis by acquiring and displaying redundant complement regulators via trogocytosis (58). Although the link between complement resistance and oxidative stress is unclear, our findings indicate that upregulated oxidative stress-related genes may contribute to complement resistance in the parasite.

To investigate the roles of targeted genes in *E. histolytica*, phenotypic assays were performed on overexpression and silencing transfectants, evaluating adherence, cytopathic activity, and phagocytosis of rice starch and red blood cells (37, 43, 59–61). CPs are key contributors to *E. histolytica* pathogenicity, with a correlation between CP activity and amoebic liver abscess formation observed in several studies (32, 62, 63). However, the assessment of CP activity and hemolytic activity, particularly in less virulent wild-type clones, did not support these findings (64, 65). Notable examples of this discrepancy are wild-type clones B1 and B12, which are non-pathogenic but exhibit the highest CP activity among all clones analyzed, suggesting that peptidase expression may vary with environmental conditions. While upregulated genes identified from RNA-seq data were associated with virulent clinical strains, our phenotypic assays using overexpression transfectants showed significant changes in adherence and phagocytosis, indicating a critical link between these processes and virulence. Particularly, our examination of pathogenicity revealed notable changes in virulence, particularly in the aspects of adherence and phagocytosis (Fig. 4). Surface molecules like Gal/GalNAc lectin mediate adhesion to host tissues, while phagocytosis facilitates host cell destruction and immune evasion, both essential for pathogenicity (9, 66). The evidence shows that trophozoite-induced mammalian cell destruction is contact dependent and triggers calcium influx. Trophozoite-induced mammalian cell destruction is contact dependent and triggers calcium influx, with adhesion mediated by surface factors like LPPG, Gal-GalNAc lectin, KERP1, and STIRP (44, 67–69). Proteins involved in phagocytosis have

been characterized, though a complete understanding of the process remains elusive. Our findings align with previous reports that multiple overlapping pathogenicity stages exist, including the role of rhomboid protease in adhesion and phagocytosis (37).

The GRIP domain in coiled-coil peripheral Golgi membrane proteins acts as a specific targeting signal for the trans-Golgi network. It includes a conserved 42-amino-acid sequence, found in animal and yeast cells (30). Although the role of GRIP-domain proteins in *E. histolytica* remains unclear, it is possible that they are associated with the trans-Golgi network to lysosomes, a system implicated in liver abscess development and resistance to nitrosative stress (70).

There are some limitations inherent in our study. First, the repressed genes in the HV group were not evaluated using the same assays as the overexpressed genes. These repressed genes are likely to play a pivotal role in maintaining the balance of pathogenesis between the host and parasite, and further research is warranted to investigate their contributions to virulent phenotypes and stress response mechanisms. Second, the functional characterization of genes in relation to virulence remains limited in the absence of loss-of-function studies as the parasites did not survive drug selection post-gene silencing, even under conditions that support cell survival. Third, overexpression assays may impact the fidelity of signal peptide processing, especially in U2 genes. Moreover, alternative translation initiation may result in truncated protein variants, leading to mislocalization in the cytoplasm, as confirmed by IFA analysis. Overexpression of U2 may further compromise *E. histolytica*'s ability to orchestrate an effective stress response, given that the parasite typically upregulates specific genes and reconfigures its metabolism under stress conditions. Such interference could impair key protective mechanisms, potentially affecting cellular homeostasis and survival. Furthermore, the overexpression model is associated with potential variability in plasmid copy number across cells, resulting in inconsistent gene expression and protein levels, thereby complicating the interpretation of findings.

Our study uncovered complex phenotypes for upregulated genes in various clinical *E. histolytica* strains involved in environmental stress, including oxidative stress and heat shock, and demonstrated a strong correlation between the stress response and virulence related to adherence and phagocytosis. Further research is required to elucidate the genes and biological significance of hypothetical proteins in different *E. histolytica* strains.

## MATERIALS AND METHODS

### Collection of clinical *E. histolytica* strains from patients with varying severity

Clinical *E. histolytica* strains were isolated from stool, intestinal fluid, and liver abscess aspirates of PCR-diagnosed patients using established cultivation protocols (19, 71, 72). Trophozoite samples were cultured in Robinson's R and R medium precultured with *Escherichia coli* (BR) media, while cyst-containing stool samples were treated with 0.1 N HCl to remove contaminants before culturing. Axenic strains were derived from monoxenic cultures using *Crithidia fasciculata* and yeast extract, iron, maltose, dipotassium phosphate, hemin, ascorbic acid, and serum (YIMDHA-S) medium. Clinical data, including symptoms, were recorded at the National Center for Global Health and Medicine, Tokyo.

### Cell culture, maintenance, and stable transfection of *E. histolytica*

Trophozoites of *E. histolytica* strains (wild type) were cultured axenically in Diamond's TYI-S-33 medium at 37°C (73). Exponential phase cells were transfected using Attractene reagent (Qiagen), following standard protocols (60, 74). After 48 h of growth, transfected parasites were selected with 2 µg/mL G418, gradually increasing to a final concentration of 24 µg/mL. Stable transformants were established based on previously described methods for *E. histolytica* (60).

## Experimental induction of amoebic abscesses in hamsters

Axenic *E. histolytica* clinical strains were cultured to the log phase (60%–80% confluence) with >90% viability confirmed via trypan blue. Trophozoites ($1 \times 10^6$) were resuspended in 100 μL BI-S-33 medium and injected into the left liver lobe of 4-week-old male Syrian hamsters (Japan SLC, Inc) (75). One week post-injection, the hamsters were euthanized, and livers with abscesses were dissected, weighed, and cultured in YIMDHA-S medium for infection confirmation. Experiments were performed in triplicate.

## RNA extraction and sequencing

Total RNA was extracted from ~$1 \times 10^6$ *E. histolytica* trophozoites (in triplicate) using a Nucleospin RNA Kit (Takara). Trophozoites were pelleted by centrifugation, lysed, and treated with RNase-free rDNase to remove genomic DNA. RNA was eluted in 50 μL nuclease-free water, quantified using a Qubit RNA BR Assay Kit (Thermo Fisher), and assessed for quality with an Agilent 2100 Bioanalyzer. The poly(A) mRNA fraction was isolated using Oligo(dT) beads and fragmented with divalent cations at high temperatures. Random primers were used to synthesize the first and second cDNA strands. Double-stranded cDNA was end-repaired, dA-tailed, and ligated with adapters using T-A ligation. Size selection of adapter-ligated DNA was performed with DNA Clean Beads, followed by PCR amplification using P5 and P7 primers. Amplified products were validated, and libraries were indexed, multiplexed, and sequenced on the Illumina HiSeq 4000 platform using a $2 \times 101$ bp paired-end configuration.

## Bioinformatic analysis of the RNA-seq data

We followed the RNA analysis workflow from our previous study (19). RNA-seq reads were trimmed and mapped to the *E. histolytica* genome (AmoebaDB v.1.7) using the CLC Genomic Workbench (Qiagen) with a gene model provided by Dr. Hon (38). Their RNA-seq data have been deposited in the European Nucleotide Archive (http://www.ebi.ac.uk/ena/data/view/ERP001024). Samples with transcript integrity numbers above 80 were selected for analysis. Orthologs among isolates were identified through AmoebaDB, and raw fragment counts were exported for statistical analysis in DESeq2. Annotated coding regions with sufficient read depth were analyzed, and data normalization was conducted using DESeq2's default parameters. Genes were considered differentially expressed if the adjusted *P* value was <0.05, following the Benjamini-Hochberg procedure and Tukey's range test. PCA was performed to explore gene expression patterns among clinical strains. Hierarchical clustering was conducted using the TCC-GUI interface, and heatmaps and volcano plots (−log10 *P* values) were generated through the CLC Genomic Workbench.

## Reverse transcriptase PCR

Reverse transcriptase PCR (RT-PCR) was performed by treating 2 μg of total RNA with DNase for 15 min at 37°C, followed by addition of EDTA (2.5 mM) and incubation at 65°C for 10 min. The RNA was split into two aliquots: one as a no-RT control and the other transcribed using SuperScript IV First-Strand Synthesis (Invitrogen) with 10 mM deoxynucleoside triphosphate at 55°C for 10 min. Two microliters of cDNA was used as a template for 30 PCR cycles. Primers used for RT-PCR are listed in Table S1. Relative gene expression was calculated by normalizing to ribosomal protein 21, which served as the internal reference. Band intensities were quantified using ImageJ analyzer software (v.1.53), and all experiments were repeated in biological triplicate to ensure reproducibility.

## Plasmid construction

The full-length coding regions of target genes were amplified by PCR from *E. histolytica* wild-type genomic DNA using each specific primer which contains SmaI and XhoI

restriction sites (Table S2). The amplified products were then cloned into a Topo TA pCR 2.1 vector (Invitrogen, USA). These wild types of target genes were subcloned into the pKT-3M vector at the SmaI and XhoI (New England Biolabs Inc., USA) restriction sites, resulting in an N-terminal triple Myc tag fusion (76). For cloning all trigger silencing constructs, the pKT-04T plasmid was employed as the backbone, with expression controlled by the cysteine synthase (CS) promoter, as previously described (31). The pKT-04T-FL-Myb plasmid was obtained from our previous study (77). All parasite plasmids are constructed from the pKT3M backbone, enabling the overexpression of proteins with an N-terminal Myc tag controlled by the CS promoter. Correct gene insertion was confirmed by sequencing.

## Western blot analysis

Western blotting was performed following standard protocols (33). Trophozoites were lysed in M-PER (Thermo Fisher) with protease inhibitors (Halt 2×, 1 mM leupeptin, phenylmethylsulfonyl fluoride, dithiothreitol, and E-64). Proteins were separated via SDS-PAGE and transferred to polyvinylidene fluoride membranes. Membranes were probed with primary antibodies, including anti-Myc (Cell Signaling; no. 2276; 1:3,000) and anti-beta-actin (Abcam; no. ab227387; 1:10,000). Secondary horseradish peroxidase-conjugated antibodies (1:10,000) were applied for 1 h at room temperature, with detection via ECL (GE). The relative protein expression normalized to β-actin, used as the internal reference, was determined by quantifying band intensities with ImageJ analyzer software (v.1.53). All experiments were conducted in biological triplicate.

## Immunofluorescence assay

The procedure was conducted as per the protocol of previous research (33, 78). Parasites were fixed in a methanol-acetone mixture (1:1) and incubated overnight at 4°C with a 1:250 dilution of mouse anti-Myc antibody. After three washes of phosphate-buffered saline (PBS), slides were incubated with a 1:1,000 dilution of Alexa 488-conjugated goat anti-mouse antibody for 1 h. The samples were then mounted in Vectashield (Vector Laboratories, Inc.) and sealed with a cover slip. Images were obtained using a Leica CTR6000 microscope with a BD CARVII confocal unit.

## Heat-shock and oxidative stress viability assay

Trophozoites were incubated at a high temperature (42°C) or in the presence of 2.5 mM hydrogen peroxide ($H_2O_2$). At 1, 2, and 3 h of incubation, samples were taken, stained with Trypan Blue (0.4% final concentration), and the number of surviving trophozoites was determined by counting them in a counting chamber under a Bausch and Lomb light microscope.

## Assay for parasite resistance to human complement

A modified protocol was used to assess parasite resistance to human complement (37, 79). Trophozoites ($1.2 \times 10^4$) were washed in PBS and incubated with 50% normal human serum (NHS) in PBS containing 0.5 mM $MgCl_2$ and 1.25 mM $CaCl_2$ for 1 h at 37°C. Controls included trophozoites treated with 50% heat-inactivated NHS or 2.5 mM $H_2O_2$. Parasites were stained with 0.2% trypan blue, and viable cells (excluding the dye) were counted. Viability with heat-inactivated NHS was set to 100%, and dead cells with $H_2O_2$ were set to 0%, with subsequent viability expressed as a percentage.

## Adhesion assay

Adherence assays with healthy CHO cells were conducted following established protocols (37, 80, 81). Approximately $1 \times 10^4$ parasites were combined with $2 \times 10^5$ CHO cells in M199 medium containing 25 mM HEPES (pH 6.8), 5.7 mM cysteine, 0.5% bovine serum albumin, and 10% heat-inactivated bovine serum. The mixture was centrifuged at

$150 \times g$ for 5 min, incubated on ice for 2 h, and subsequently vortexed briefly. Parasites with three or more attached CHO cells were considered adhesion positive and counted using a hemocytometer.

## Cytopathic assay

As described in previous methods (26, 61), $6 \times 10^4$ trophozoites were applied to a confluent monolayer of CHO cells and incubated at 37°C for 2 h. To remove the trophozoites, the culture was placed on ice for 15 min. Cells were washed three times with cold PBS supplemented with 2% (vol/vol) galactose, fixed with 4% ultra-pure formaldehyde for 15 min, and stained using 0.1% methylene blue (OmniPur, USA) in 10 mM borate buffer (pH 8.7). After three washes, dye extraction was performed using 500 μL of 0.1 M HCl, incubated at 37°C for 30 min. The extract was diluted 1:10 with distilled water, and absorbance at 645 nm was recorded. The undamaged monolayer was considered 0% destruction, while the syringe-lysed control represented 100% destruction.

## Phagocytosis assay

Erythrophagocytosis assays were conducted following a previously published protocol (37). In brief, $5 \times 10^6$ human RBCs were incubated with $1 \times 10^5$ trophozoites in 400 μL of culture medium at 37°C for 30 min. Ice-cold distilled water was applied to lyse any remaining extracellular RBCs, followed by two rounds of centrifugation. The resulting pellet, containing parasites that had ingested RBCs, was lysed using 90% formic acid, and absorbance at 405 nm was measured spectrophotometrically. For evaluating starch ingestion, $2.5 \times 10^5$ trophozoites were seeded in 24-well plates and allowed to adhere for 2 h at 37°C. These were then exposed to TYI-S-33 medium with 0.004% rice starch for 1 h. After washing with PBS, cells were fixed with 4% formaldehyde for 20 min, permeabilized in 70% ethanol, and stained with 1% Lugol's solution for 10 min at room temperature. Dark brown-stained rice starch grains indicated ingestion, with trophozoites containing at least one grain considered positive for phagocytosis. A total of 100 parasites were counted for each condition, and the assay was performed three times.

## Statistical analysis

All data were processed and analyzed using GraphPad Prism (v.8.0.1) software. Comparisons between samples and controls were performed using two-way Student's $t$-test, depending on the experimental design. Results are presented as the mean ± standard deviation from at least three independent biological experiments, each conducted with two technical replicates.

## ACKNOWLEDGMENTS

We are grateful to Hangbang Zhang and Daniela Lozano-Amado for helpful discussion and advice. A grant from the Uehara Memorial Foundation was provided to Y.Y. (no grant number applied).

U.S. was funded by the National Institutes of Health and supervised the project. S.M. conceived of the overall approach and performed the immunofluorescence assay. S.I. performed and analyzed the RNA-seq data. Y.Y. designed, performed, and analyzed the experiments and wrote the manuscript.

## AUTHOR AFFILIATIONS

[1]Department of Microbiology and Immunology, Stanford University School of Medicine, Stanford, California, USA
[2]AIDS Clinical Center, National Center for Global Health and Medicine, Tokyo, Japan
[3]Department of Parasitology, National Institutes of Infectious Diseases, Shinjuku, Tokyo, Japan

⁴Division of Infectious Diseases, Department of Internal Medicine, Stanford University School of Medicine, Stanford, California, USA

## AUTHOR ORCIDs

Yasuaki Yanagawa  http://orcid.org/0000-0002-3723-2425
Manu Sharma  http://orcid.org/0000-0003-3120-9854
Upinder Singh  http://orcid.org/0000-0003-0630-0306

## FUNDING

| Funder | Grant(s) | Author(s) |
|---|---|---|
| Uehara Memorial Foundation | | Yasuaki Yanagawa |

## AUTHOR CONTRIBUTIONS

Yasuaki Yanagawa, Conceptualization, Data curation, Formal analysis, Investigation, Methodology, Software, Visualization, Writing – original draft | Manu Sharma, Investigation, Methodology | Shinji Izumiyama, Data curation, Methodology | Upinder Singh, Funding acquisition, Project administration, Supervision

## DATA AVAILABILITY

The sequencing data have been deposited in the DNA Data Bank of Japan (https://www.ddbj.nig.ac.jp/index-e.html) under accession number E-GEAD-878.

## ETHIC APPROVAL

This study was approved by the National Center for Global Health and Medicine ethics committee (NCGM-G-001566-02) and conducted per the Declaration of Helsinki guidelines. Animal procedures were approved by the National Institutes of Infectious Diseases ethics committee (117155-IV) and complied with the Ministry of the Environment's standards for laboratory animal care and pain relief.

## ADDITIONAL FILES

The following material is available online.

### Supplemental Material

**Supplemental figures (Spectrum00506-25-s0001.pdf).** Fig. S1 and S2.
**Supplemental tables (Spectrum00506-25-s0002.pdf).** Tables S1 and S2.

### Open Peer Review

**PEER REVIEW HISTORY (review-history.pdf).** An accounting of the reviewer comments and feedback.

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
