## [Reviewer comments · Microbiology Spectrum]

Microbiology Spectrum

Exploring Virulence and Stress Response in *Entamoeba histolytica*: Insights from Clinical Strains

Yasuaki Yanagawa, Manu Sharma, Shinji Izumiyama, and Upinder Singh

Corresponding Author(s): Upinder Singh, Stanford University

Review Timeline:

Submission Date:	February 25, 2025
Editorial Decision:	March 24, 2025
Revision Received:	April 16, 2025
Accepted:	May 6, 2025

Editor: Prakash Srinivasan

Reviewer(s): The reviewers have opted to remain anonymous.

Transaction Report:

DOI: <https://doi.org/10.1128/spectrum.00506-25>

Re: Spectrum00506-25 (Exploring Virulence and Stress Response in *Entamoeba histolytica*: Insights from Clinical Strains)

Dear Dr. Upinder Singh:

Thank you for the privilege of reviewing your work. Below you will find my comments, instructions from the Spectrum editorial office, and the reviewer comments.

Several major concerns to prior reviews have not been sufficiently addressed. For example this includes insufficiently addressing reviewer 1's point 4 about using a normalization control amplified using cDNA and not gDNA. Also, the error bars shown in the quantitation, do they represent biological replicates or technical replicates and how many replicates. Likewise, for western blot quantitation, the use of actin from WT lane that does not have any myc tagged protein

Reviewer 1's point 5 regarding the the tag placement ahead of the signal sequence and its impact on expression (Fig 2c) localization and function (Fig 3A) needs clarification. Likewise, the use of a more suitable control to normalize for Myc-tagged proteins is warranted.

As the reviewer questions are regarding the technical aspects of the data, these need to be addressed before it can be considered further.

Revision Guidelines

Sincerely,
Prakash Srinivasan

Re: Spectrum00506-25R1 (Exploring Virulence and Stress Response in *Entamoeba histolytica*: Insights from Clinical Strains)

Dear Dr. Upinder Singh:

I am happy to inform you that your manuscript has been accepted, and I am forwarding it to the ASM production staff for publication. Your paper will first be checked to make sure all elements meet the technical requirements. ASM staff will contact you if anything needs to be revised before copyediting and production can begin. Otherwise, you will be notified when your proofs are ready to be viewed.

Sincerely,
Prakash Srinivasan
Editor
Microbiology Spectrum